# Development of Two Patient Self-Reported Measures on Functional Health Status (FOD) and Health-Related Quality of Life (QOD) in Adults with Oropharyngeal Dysphagia Using the Delphi Technique

**DOI:** 10.3390/jcm11195920

**Published:** 2022-10-07

**Authors:** Renée Speyer, Reinie Cordier, Deborah Denman, Catriona Windsor, Gintas P. Krisciunas, David Smithard, Bas J. Heijnen

**Affiliations:** 1Department Special Needs Education, University of Oslo, 0318 Oslo, Norway; 2Curtin School of Allied Health, Faculty of Health Sciences, Curtin University, Perth, WA 6102, Australia; 3Department of Otorhinolaryngology and Head and Neck Surgery, Leiden University Medical Center, 1233 ZA Leiden, The Netherlands; 4Department of Social Work, Education and Community Wellbeing, Faculty of Health & Life Sciences, Northumbria University, Newcastle upon the Tyne NE7 7XA, UK; 5Department of Linguistics, Faculty of Medicine, Health, and Human Sciences, Macquarie University, Sydney, NSW 2109, Australia; 6School of Medicine, Boston University, Boston, MA 02118, USA; 7Centre for Exercise and Active Rehabilitation, University of Greenwich, London SE9 2BP, UK

**Keywords:** swallowing disorders, deglutition, questionnaire, survey, content validity, instrument development, consensus, definition, self-evaluation, self-report

## Abstract

*Introduction*. Patient self-evaluation is an important aspect in the assessment of dysphagia and comprises both Functional Health Status (FHS) and Health-Related Quality of Life (HR-QoL). As many measures combine both FHS and HR-QoL, disease-related functioning cannot be distinguished from disease-related quality of life as experienced by the patient. Moreover, current patient self-reported measures are limited by poor and incomplete data on psychometric properties. *Objective*. This study aimed to establish content validity for the development of two new self-reported measures on FHS and HR-QoL in adults with oropharyngeal dysphagia (OD), in line with the psychometric taxonomy and guidelines from the COSMIN group (**CO**nsensus-based **S**tandards for the selection of health **M**easurement **IN**struments). *Methods*. Using the Delphi technique, international expert consensus was achieved; participants and patients with dysphagia evaluated relevance, comprehensiveness, and comprehensibility of definitions of relevant constructs (i.e., dysphagia, FHS and HR-QoL) and potential items. *Results*. A total of 66 Delphi participants from 45 countries achieved consensus across two rounds. The Delphi study resulted in two prototype measures, the Functional health status measure of Oropharyngeal Dysphagia (FOD) and the health-related Quality of life measure of Oropharyngeal Dysphagia (QOD), consisting of 37 and 25 items, respectively. Minimal revisions were required based on feedback by patients. *Conclusions*. This study provides evidence of good content validity for both newly developed prototype measures FOD and QOD. Future studies will continue the process of refining the measures, and evaluate the remaining psychometric properties using both Classic Test Theory (CTT) and Item Response Theory (IRT) models.

## 1. Introduction

Problems in swallowing or oropharyngeal dysphagia (OD) is associated with high morbidity and mortality rates [1]. OD can lead to dehydration, malnutrition and even death, and may have a major impact on a patient’s health-related quality of life and well-being [2]. While most clinicians and researchers agree on the impact and consequences of OD as well as on the need to intervene, there is still disagreement on definitions of OD and choices in screening and assessment [3].

Patient self-evaluation is considered an important aspect in the repertoire of OD assessments. Self-evaluation assessment of OD comprise broadly two aspects, Functional Health Status (FHS) and Health-Related Quality of Life (HR-QoL). FHS refers to the influence of a given disease (e.g., OD) on particular functional aspects, whereas HRQOL refers to the unique personal perception of someone’s health, taking into account social, functional, and psychological issues [4]. Even though considered two distinct concepts, many questionnaires combine both FHS and HR-QoL. Consequently, disease-related functioning cannot be distinguished from disease-related quality of life as experienced by the patient. Further, two psychometric reviews on patient self-evaluation in OD (i.e., Speyer, Cordier [5], Timmerman, Speyer [6]) reported that most measures did not evaluate all psychometric properties and poor psychometric quality for those psychometric properties that were evaluated. Moreover, the findings from the systematic reviews indicate that content validity for current measures are lacking, indicating that measure measures lack a sound conceptual framework. The findings highlight an urgent need for more research and development of new instruments using and reporting on pre-established criteria as recommended in literature.

The COSMIN group (**CO**nsensus-based **S**tandards for the selection of health **M**easurement **IN**struments) established an international consensus-based taxonomy, terminology and definitions of measurement properties for health-related patient-reported outcomes [7,8]. The framework comprises nine measurement properties within three domains: reliability, validity and responsiveness. According to the COSMIN framework, content validity is the most important measurement property of a patient-reported measure, and refers to the degree to which the content of an instrument is an adequate reflection of the construct to be measured [7]. If content validity is flawed or lacking, the measure is of questionable value for either clinics or research.

To meet the COSMIN standards for good instrument development, a measure should be developed using recent literature on the topic of interest and involve clinical experts and targeted patient groups. Content validity should be assessed by asking both professionals and patients about the relevance (all items should be relevant to the construct of interest within a specific population and context of use), comprehensiveness (no key aspects of the construct should be missing), and comprehensibility (the items should be understood by patients as intended) of the items of a measure [7].

Informed by the COSMIN guidelines [7], this study reports on the first step towards the development of two patient self-report measures on FHS and HR-QoL in dysphagia: the Functional health status measure of Oropharyngeal Dysphagia (FOD) and the Quality of life measure of Oropharyngeal Dysphagia (QOD). To support good content validity, this study reports on: (1) an international Delphi study involving dysphagia experts to seek agreement on definitions and items of two prototype measures on FHS and HR-QoL; and (2) patients’ feedback on the relevance, comprehensiveness and comprehensibility of the items of both prototype measures.

## 2. Methods

### 2.1. Study Design

This study used the Delphi technique which is a structured process that aims to develop group consensus on a defined topic through a series of survey rounds [9]. Consecutive surveys are modified, informed by percentage of agreement and feedback received in preceding rounds. Participants are experts in a specific content area and remain anonymous from each other across rounds to avoid bias and some Delphi participants dominating the consensus process [9]. The same participants complete each round, even though some participants may not participate in all Delphi rounds. Delphi rounds continue until consensus has been reached or it becomes apparent that consensus cannot be reached. In this study, online surveys (e-Delphi) were used to seek expert consensus regarding the constructs and items to be included in patient self-report measures on FHS and HR-QoL in OD. Following the Delphi study, patients with OD were recruited and asked about the relevance, comprehensiveness and comprehensibility of all items of both prototype measures. Patients’ feedback was used to revise both preliminary measures.

### 2.2. Participants

To be eligibility to participate in the Delphi study participants needed to: (1) have English reading skills adequately for work (e.g., understanding the main points of texts and technical terms within participant’s field of expertise); (2) have five or more years clinical, research or teaching experience where at least 20% or more of the caseload related to adults with dysphagia (e.g., provision of clinical services, research, and/or staff development and academic teaching); and (3) engaged in activities related to dysphagia for at least one full day per week over the past two years.

### 2.3. Procedure

#### 2.3.1. Recruitment

The study was approved by the Curtin Human Research Ethics Committee (Curtin University, Perth, Australia: HRE2018-0627). Delphi participants were recruited via professional organisations (e.g., European Society for Swallowing Disorders, Dysphagia Research Society, Royal Collage of Speech Language Therapists, Speech-Language & Audiology Canada, South African Speech-Language-Hearing Association, New Zealand Speech-Language Therapists’ Association), from the professional networks of the authors, and by asking recruited participants to identify other potential participants (snowballing). Identified participants were sent an invitation and a participant information sheet about the Delphi study. Participants who accepted the invitation were sent a link to the first online Delphi survey. Participants who did not complete the first survey round were excluded from the subsequent rounds. Patients with OD were recruited via authors’ clinical networks. Patients were invited during outpatient visits.

#### 2.3.2. e-Delphi Surveys

Based on international literature and existing patient self-reported questionnaires on FHS and HR-QoL as identified in previous reviews [5,6], definitions for main concepts (i.e., dysphagia, FHS, HR-QoL) and lists of potential measure items were constructed by the first and second author. Both definitions and potential items were presented to participants across two rounds via an online survey platform (www.qualtrics.com) over a six-month period (May–October 2020). Participants indicated agreement on definitions using 5-point ordinal scales (i.e., Strongly agree, Agree, Neither agree nor disagree, Disagree, Strongly disagree), by ranking the most preferred definitions and providing suggestions for rephrasing or alternative definitions. In addition, participants rated the importance of retaining items (i.e., relevance [COSMIN]) using an ordinal 5-point scale (i.e., Essential [always assess], Important [assess in most situations], Limited [assess in some situations], Irrelevant [inappropriate to assess], Unsure). They were able to add comments about items or make suggestions for alternative wording of each item separately (i.e., comprehensibility [COSMIN]). Responses were analysed after each Delphi round by two authors, and definitions and items were revised according to participants’ comments. For each round, an open-ended comment section was available to add items that may be missing in fully capturing the underlying constructs (i.e., comprehensiveness [COSMIN]).

Definitions, FHS items and HR-QoL items were presented as separate survey parts. To ensure completion time of the Delphi survey was feasible, HR-QoL items were not included in the first Delphi round and only introduced in the second round. Between Delphi rounds, participants received summarised findings, including information on participants characteristics, percentage agreement on definitions and items, and revisions were made using participants’ feedback (i.e., rewording definitions, revising, and adding or deleting items).

#### 2.3.3. Patient Questionnaire

A questionnaire was developed including all items of both preliminary measures for FHS and HR-QoL resulting from the Delphi study. Patients were asked to rate the importance (i.e., relevance [COSMIN]) and ease of understanding (i.e., comprehensibility [COSMIN]) of each item using a 5-point rating scale (i.e., Strongly agree, Agree, Neither agree nor disagree, Disagree, Strongly disagree; converted scores 5–1). Patients were able to suggest alternative wording for each item using open boxes that were provided throughout the questionnaire. At the end of each measure, patients were asked about any missing construct of interest that was not captured by the listed items (i.e., comprehensiveness [COSMIN]). Patients could opt to complete the questionnaire independently in the waiting room or at home, or during an onsite or online semi-structured interview in the presence of one of the researchers.

### 2.4. Analysis

Survey responses were analysed using the Statistical Package for the Social Sciences [10]. Criteria for agreed consensus were defined prior to the study; consensus between participants was achieved if at least 75% of respondents indicated ‘Strongly agree’ or ‘Agree’ for definitions, and ‘Essential [i.e., always assess]’ or ‘Important [i.e., assess in most situations]’ for items [11,12]. The total number of Delphi rounds were to be determined by level of agreement following each round.

Participants’ responses to open-ended questions were analysed per item and categorised by constructs (i.e., FHS and HR-QoL), before deciding on the rewording of original items or creating a new item based on participants’ suggestions. Data analysis was performed by the first and second author who were blinded to the identity of participants.

## 3. Results

### 3.1. Delphi Participants

A total of 94 participants agreed to participate in the Delphi study, of which three invitees did not complete the survey. Of the 91 participants, one did not meet the eligibility criterion ‘being engaged in activities related to dysphagia for at least one full day per week over the past two years’ and twelve did not meet the eligibility criterion ‘five or more years clinical, research or teaching experience where at least 20% or more of the caseload related to adults with dysphagia’. The second Delphi round was completed by 66 participants (84.6%) of the 78 participants who also completed the first Delphi round. As participants achieved consensus on all definitions and items within two Delphi rounds, no further rounds were needed. Appendix A presents demographics of all participants who completed the first (*n* = 78) and second Delphi rounds (*n* = 66).

Of the participants from the second Delphi round, the majority resided in Europe; 74% of participants originated from 18 different European countries, whereas 26% originated from outside of Europe (27 countries from four continents). Experts’ highest qualification related to the field of dysphagia was a PhD in 50% of participants, a Master degree in 36% of participants, and a Bachelor degree in 14% of participants. Participant’s professional backgrounds varied; 62% were speech and language pathologist, 26% were medical specialist, and 9% were either occupational therapists or dieticians. Two experts (3%) were trained both as allied healthcare professionals and medical specialists. Participants worked in one or more different practice settings; over 70% of participants (71%) were working in a hospital, 33% in education (e.g., university), 15% in private practices, 12% in community health centres, and 3% in residential aged care or disability care. Eight percent were students. Participants worked with different dysphagia patient populations. Of the main dysphagia populations that the second round Delphi participants (*n* = 66) worked with, 81% worked with people with non-degenerative or acquired neurological trauma, 79% work with people with degenerative neurological disorders, 42% worked with elderly in geriatric care, 52% worked with patients with oncological diseases, 27% worked with patients with respiratory diseases, and 21% worked with patients with gastrointestinal diseases. All participants have engaged in activities related to adults with dysphagia (e.g., provision of clinical services, research, and/or academic teaching) for at least five years: in the first Delphi round, 22% of experts had 5–10 years of experience; 27% between 11–15 years; 18% between 16–20 years; 24% between 21–30 years; and 9% over 30 years of experience. In the second Delphi round, 23% of experts had 5–10 years of experience; 30% between 11–15 years; 17% between 16–20 years; 21% between 21–30 years; and 9% over 30 years of experience.

### 3.2. Delphi Process

The first Delphi round included definitions for two main concepts (i.e., dysphagia and FHS) and 43 FHS-related items. The second Delphi round included rephrased and new definitions for the concepts of dysphagia and FHS and the concept of HR-QoL was added. In terms of items, 8 items were reworded and 9 new FHS items were added as suggested by participants during the first round. In addition, 34 HR-QoL items were introduced to Delphi round 2. Detailed decision trees for definitions on dysphagia, FHS and HR-QoL is provided in Table 1, and for FHS and HR-QoL items in Table 2.

### 3.3. Definitions for Dysphagia, FHS and HR-QoL

During the first Delphi round, participants were presented with three definitions for dysphagia and four for FHS. Participants indicated consensus per definition by rating definitions according to preference, and provided suggestions for rewording. In the second Delphi round, the highest ranked definitions for dysphagia and FHS fol-lowing round one were compared with two new definitions based on participants’ feedback. In Delphi round two, participants were also asked to rate agreement and rank three definitions for HR-QoL, and provide feedback for rephrasing. Table 1 pre-sents the Delphi process for establishing consensus for all three concepts in two Delphi rounds.

### 3.4. FHS and HR-QoL Items

*Delphi round I.* Agreement was reached for 33 items out of the original 43 FHS items, with percentage agreement set at 75% and above. Based on expert feedback, one item was split into two separate items. For the second Delphi round, six items were rephrased; one of which was compared with the original item to obtain preference ranking. Three items approached 75% agreement, of which one item was covered by the constructs of the 33 items for which consensus was already achieved. Based upon participants’ suggestions for revisions, the other two items were reworded and also added to the second Delphi round. Seven items for which participants did not achieve agreement (percentage agreement ≤ 70%) were deleted. In summary, 27 items were accepted without revision, eight items were deleted, and seven items were reworded to be included in Delphi round II (see Table 2). One item was rephrased for comparison with the original item. In addition, based on recommendations following prompts to indicate comprehensiveness of items, nine new items were added.

*Delphi round II.* The second Delphi round consisted of 17 FHS items resulting from the first round, and 34 new HR-QoL items. For eight of the 17 FHS items, participants achieved agreement of 75% and above (including the rephrased comparison item) and were therefore accepted. The remaining nine items were deleted. In addition, one item originally included under HR-QoL was added to the FHS items. Finally, after two Delphi rounds, the prototype measure on FHS included a total of 37 items (see Table 2).

Using the cut-off of 75% and above agreement, 26 out of 34 items on HR-QoL were retained. As one of these items was actually targeting FHS, it was moved to the FHS measure. Using participants’ comments, four of the included items were slightly reworded to improve comprehensibility. As a result, the final prototype measure on HR-QoL included 25 items.

Further details on the process of FHS and HR-QoL item selection during both Delphi rounds, is reported in Table 2. The table presents the decision tree for item agreement and processes around decision-making, resulting in two prototype measures, the FOD (37 items) and the QOD (25 items), respectively. Example items of both prototype measures can be found in the Appendix A.

### 3.5. Patient Stakeholder Group

Seven patients with confirmed OD by instrumental assessment (videofluoroscopic or endoscopic evaluation of swallowing) completed the patient questionnaire: four males and 3 females, having an averaged mean age of 70 years (range: 61–82). Patient’s medical diagnoses comprised head and neck cancer (*n* = 4), neurological diseases (*n* = 2) and sarcoidosis (*n* = 1). Patients represented a wide range of dysphagia severity as reflected in scores on the Functional Oral Intake Scale (FOIS; Crary, Mann [22]) and the Penetration Aspiration Scale (PAS; Rosenbek, Robbins [23]). FOIS is a 7-point ordinal scale (1 = Nothing by mouth—7 = Total oral diet with no restriction) and PAS is an 8-point ordinal, visuoperceptual scale used to evaluate instrumental recordings of swallowing (1 = Material does not enter the airway—8 = Material enters the airway, passes below the vocal folds, and no effort is made to eject). Mean scores on FOIS and PAS were respectively 3.7 (range: 1–7) and 5.8 (range: 1–8).

For most items, patients showed high agreement for both relevance and comprehensibility: both mean relevance and comprehensiveness scores indicated ‘Agree’ (mean converted relevance scores = 3.8, range: 2.7–4.9; mean converted comprehensiveness scores = 4.1, range: 2.9–4.7). All patients confirmed that both measures were comprehensive having no key aspects missing. Based on patients’ feedback minor revisions were made in the wording of four items.

## 4. Discussion

### 4.1. Content Validity

This current study is the first step towards the development of two patient self-report measures targeting FHS and HR-QoL in dysphagia: the FOD and the QOD. To meet COSMIN standards for content validity as being part of good instrument development, an international Delphi study amongst dysphagia experts was conducted to seek agreement on definitions of relevant concepts (i.e., dysphagia, FHS and HR-QoL), and on the relevance, comprehensiveness, and comprehensibility of items of both prototype measures [7].

An initial 78 experts from 31 countries over five continents participated in this e-Delphi, representing a large collection of health professionals and disciplines involved in dysphagia care. Over 80% of participants had a higher degree of research in dysphagia, of which over 50% have completed a PhD. In addition, about 80% of experts reported having over ten years of experience in dysphagia care, and almost 35% having over twenty years of experience. Therefore, the e-Delphi included an international, highly qualified and experienced dysphagia expert group, meeting the highest standards for soliciting professionals’ opinions according to COSMIN criteria [7]. Further, with 66 participants having completed both Delphi rounds and a retention rate of 84.6%, it exceeded the minimum of 50 participants as recommended by the COSMIN guidelines [7] to achieve high methodological quality. Similarly, the involvement of a group of patient stakeholders (*n* = 7) providing feedback on the relevance, comprehensiveness, and comprehensibility of the items of both prototype measures, met the COSMIN recommendations on establishing good content validity.

### 4.2. Instrument Development

The terms dysphagia and swallowing disorders are used interchangeably in the literature, without having universal consensus on how to define dysphagia [3]. The current Delphi study is the first study aiming to achieve international expert consensus on how to define dysphagia. In addition, both FHS and HR-QoL were defined: two different concepts, frequently combined in outcome measurements, conflating the distinction between disease-related functioning and disease-related quality of life as experienced by the patient. Participants’ levels of agreement ranged between 88–95% for all three definitions, indicating high consensus overall. For the final 62 included FHS and HR-QoL items, the level of agreement between participants was 75% or above. Overall, item agreement was good, and patient feedback resulted in only minor rephrasing of four items.

### 4.3. Psychometrics

Achieving international consensus on main concepts and deciding on eligibility of items based on literature and experts’ feedback, is the first step in the process of instrument development. The final prototype measures for FHS and HR-QoL, consist of 37 (FOD) and 25 (QOD) items, respectively. Both measures will be trialled with larger groups of patients with dysphagia to confirm the relevance and the comprehensibility of items, and comprehensiveness of each measure, as the next step in the process of instrument development. Psychometric properties will be determined using both Classic Test Theory (CTT) and Item Response Theory (IRT; Rasch analyses).

Although methodologies and interpretation of CTT findings are easier to interpret than IRT results, the CTT framework has limitations. The CTT framework evaluates the performance of a measure as a whole, and psychometric data are specific to the sample populations used to evaluate the measure. By contrast, IRT evaluates the reliability of each individual item, and neither unit of analysis nor results are restricted to the test population [24,25,26]. IRT models estimate item and person parameters within the same model, determine person-free parameter estimation and item-free trait level estimation, and identify optimal scaling of individual differences based on differential item functioning [26].

The COSMIN taxonomy will be used as guideline for evaluation of the psychometric properties of both preliminary measures in FHS and HR-QoL [8]. The current Delphi study and patient questionnaire confirmed good content validity [7]. The internal structure of each measure will be investigated by evaluating structural validity and internal consistency. Next, reliability, measurement error and hypotheses testing for construct validity (e.g., convergent validity) will be determined. Using both measures for pre- and post-intervention assessment in patients with dysphagia, responsiveness or the ability to detect change over time, will finally be evaluated. As no ‘gold standard’ assessment is available for patient self-reported assessment of FHS and HR-QoL in dysphagia, criterion validity cannot be determined. Further, cross-cultural validity will only be considered once a comprehensive psychometric evaluation is undertaken and a need for translation into other languages are expressed. Finally, although not considered psychometric properties, feasibility and interpretability of both measures will be evaluated within both clinical and research settings, by assigning qualitative or clinical meaning to quantitative scores.

### 4.4. Limitations

Participants in this Delphi study represented a variety of geographical locations and experts from different health care professional backgrounds. Still, inherent to Delphi studies, results may differ depending on the participants included. Also, participant drop-out over Delphi rounds poses limitations on the interpretation of results, even though completion rate for the second round was 84.6%, which is acceptable for web-based surveys having overall response rates typically of 80% or higher [27]. Furthermore, even though the patient stakeholder group comprised of a representative group of seven adult patients with OD, outcomes might have been different if other subjects have been included.

## 5. Conclusions

This study reports on the first steps towards the development of two patient self-report measures on FHS and HR-QoL in dysphagia: the FOD and the QOD. Following COSMIN guidelines, an international Delphi study among dysphagia experts resulted in two prototype measures for FHS and HR-QoL, consisting of 37 and 25 items, respectively. The current Delphi study combined with the results from patients’ feedback on relevance, comprehensiveness, and comprehensibility of the items of both measures, indicated good content validity. Future studies will continue the process of instrument development, and determine its psychometric properties using both CTT and IRT models.

## Figures and Tables

**Table 1 jcm-11-05920-t001:** Decision flow chart for definitions on dysphagia, FHS and HR-QoL: Delphi Round I and II.

Concept	Definitions ^a^	Delphi Round I			Delphi Round II		
		Level of Agreement ^b^	Preferred Definition ^c^	Decision	Level of Agreement ^b^	Preferred Definition ^c^	Decision
* **Dysphagia** *	(1) Dysphagia is defined as disordered movement of the bolus from mouth to stomach due to abnormalities in the structures critical to swallowing or in their movements [13].	74.0%	19.5%	Exclude	N/A	N/A	N/A
	(2) Dysphagia is defined as dysfunction or impairment of the stages of swallowing. It is defined by its clinical features because it is a symptom, or a collection of symptoms of one of a number of possible underlying disorders. In patients with dysphagia, various aspects of the anatomy or neurophysiology in any or all of these stages may be impaired, resulting in the diagnosis of a swallowing disorder [14].	* **90.9%** *	* **44.2%** *	* **Move to round II ^d^** *	87.9%	28.8%	Exclude
	(3) Dysphagia is an impairment of emotional, cognitive, sensory, and/or motor acts involved with transferring a substance from the mouth to stomach, resulting in failure to maintain hydration and nutrition, and posing a risk of choking and aspiration [15].	74.1%	36.4%	Exclude	N/A	N/A	N/A
	**(4)** **Dysphagia is a symptom or a collection of symptoms of one or more underlying anatomical abnormalities, or impairments and disorders in cognitive, sensory and motor acts involved with transferring a substance (including food and liquids) from the mouth—or nasal cavity—to the stomach, possibly resulting in but not limited to: reduced efficiency and safety of swallowing, failure to maintain hydration and nutrition, risk of choking and aspiration leading to pulmonary complications, and reduced quality of life.**	N/A	N/A	***New definition*** (feedback participants)	* **87.9%** *	* **71.2%** *	* **Final consensus** *
* **FHS** *	(1) The influence of a given disease on particular functional aspects and aims to quantifying the symptomatic severity of that disease as experienced by the patient [4].	87%	22.1%	Exclude	N/A	N/A	N/A
	(2) Individual’s ability to perform normal daily activities required to meet basic needs, fulfil usual roles and maintain health and well-being [16,17].	89.6%	14.3%	Exclude	N/A	N/A	N/A
	(3) Description of body functions, activities and participation in presence of a disease or infirmity of an individual or population at a particular point in time against identifiable standards [18,19].	76.6%	16.9%	Exclude	N/A	N/A	N/A
	(4) The ability to perform tasks in multiple domains (physical, social, role, and psychological functioning) and measures the focus on (loss of) function due to disease and/or treatment and the effects on daily life [6,17].	*97.4%*	*46.8%*	* **Move to round II ^d^** *	90.9%	37.9%	Exclude
	(5) * **FHS is the impact of a given disease on the ability to perform tasks in multiple domains (including physical, social, role and psychological functioning), and aims to quantify the symptomatic severity and (loss of) function due to that disease and/or treatment and the effects on daily life as experienced by the individual at a particular point in time.** *	N/A	N/A	***New definition*** (feedback participants)	* **95.4%** *	* **62.1%** *	* **Final consensus** *
* **HR-QoL** *	(1) ***HR-QoL refers to the unique personal perception of someone’s health, taking into account social, functional and psychological issues*** [4].	N/A	N/A	N/A	* **94.0%** *	* **48.5%** *	* **Final consensus** *
	(2) HR-QoL is defined as physical, psychological, and social effect on different areas of health status which changes according to the person’s experiences, beliefs, expectations, and perception [20].	N/A	N/A	N/A	86.4%	45.5%	Exclude
	(3) HR-QoL is the value assigned to duration of life as modified by the impairments, functional states, perceptions, and social opportunities that are influenced by disease, injury, treatment or policy [21].	N/A	N/A	N/A	39.4%	6.1%	Exclude

^a^ Definitions in bold-italic represent the final wording per concept as by participants’ consensus. ^b^ Consensus between participants was achieved if ≥75% of respondents indicated ‘Strongly agree’ or ‘Agree’. ^c^ Most preferred definition. ^d^ Highest level of agreement AND most preferred definition, therefore moving to Round II. *Note*. N/A = not applicable; FHS = Functional Health Status; HR-QoL = Health-Related Quality of Life.

**Table 2 jcm-11-05920-t002:** Decision flow chart for items on FHS and HR-QoL: Delphi Round I and II.

Construct	Round I			Round II			
	Total Number of Items	Agreement	Decision	Total Number of Items	Agreement	Decision	Final Number of Items
*FHS*	43 items	≥75% agreement: 33 items	Accepted items: 26 Accepted item (split into 2 separate items): 1 Reworded items: 5 ^a^ Comparison (original/rewarded item): 1 ^a^	17 items	≥75% agreement: 8 items (including comparison: 1 item) ^b^	Accepted items: 7 Ranking (original/reworded item): 1 ^c^	Round I: 28 items agreement Round II: 9 items agreement *37 items* ^f^
		<75% agreement: 10 items	Reworded items: 2 ^a^ Deleted items: 8		<75% agreement: 9 items	Deleted items: 9	
		N/A	New items (feedback participants): 9 ^a^		N/A	Extra item (moved from HR-QoL): 1 ^d^	
*HR-QoL*	N/A			34 items	≥75%: agreement 26 items	Accepted items: 21 Reworded items: 4 ^e^ Moved item to FHS: 1 ^d^	Round II: 25 items agreement *25 items* ^f^
					<75% agreement: 8 items	Deleted items: 8	

^a^ Items included in Delphi Round II. ^b^ After participants’ feedback (Round I), one item was rephrased and compared with the original item (Round II). ^c^ Participants’ feedback (Round II) indicated preference for the original item (Round I). ^d^ After participants’ feedback (Round I), one item originally listed within the construct of HR-QoL (mismatch) was moved to the FHS construct (Round II). ^e^ After participants’ feedback (Round II), four items were slightly reworded to improve comprehensibility. ^f^ Numbers in bold-italic represent the final number of items included as by participants’ consensus. *Note*. N/A: not applicable.

## Data Availability

The data presented in this study are available on request from the first author.

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
