# Peer review of "Development of Two Patient Self-Reported Measures on Functional Health Status (FOD) and Health-Related Quality of Life (QOD) in Adults with Oropharyngeal Dysphagia Using the Delphi Technique"

_jcm, 2022, doi:10.3390/jcm11195920_

Round 1

Reviewer 1 Report

Dear authors,

I congratulate you for the effort to propose new PROMS in oropharyngeal dysphagia strictly following a guideline (COSMIN). The article has a clear purpose, but I have some comments and doubts. See below:

- Page 2, line 56-58: the cited systematic reviews are from 2014. Therefore, I suggest that you do not refer to them as "recent". In almost ten years this scenario could have changed. I also suggest that you mention if there has been an update of these systematic reviews.

- Page 2, lines 74. According to the authors, "a measure should be developed using recent literature on the topic of interest and involve clinical experts and targeted patient groups". However, this study does not clearly mention how the recent literature was used. On page 3, lines 126-128, the authors used references 5 and 6 as sources for definitions and item lists. However, the references are almost ten years old.

- Page 3, items 125-142. Were the experts informed by the percentage of agreement between the first and second rounds? I suggest the inclusion of this information in item 2.3.2.

- I understand that both FHS and HR-QoL definitions and potential items were mixed to the experts and patients. Could the authors justify the reasons for not presenting the two PROMS separately? 

- Page 5, lines 181-204. There is a lot of information in the text that was also in Table 1. I think that paragraph should be reduced and just comment on the results of Table1, but not repeat them.

- The HR-QoL concept and items were only added in the second round. Why? It must be declared in the Method section.

- Page 8, Table 3: a "%" symbol is missed next to FHS, Round I, agreement (70 < agreement).

- Page 9, sub-item 3.5: what is the criteria for deciding the number of participating patients? There is a difference between the number of specialists and the number of patients. It seems that patients from different countries was not a criteria, for example.

Reviewer 2 Report

This manuscript summarizes the most extensive work done to date to develop tools with high content validity for assessing functional health status and health-related quality of life (HRQOL) in adults with oropharyngeal dysphagia.

In order to develop comprehensive questionnaires with the most relevant items, the authors enlisted the participation of 78 experts in oropharyngeal dysphagia from different countries. All of these experts had more than five years of experience; more than half had a PhD; they came from a variety of clinical-practice backgrounds; all possessed enough experience with patients to be familiar with most causes of dysphagia. To also help develop the questionnaires, the authors enlisted the participation of patients with a wide range of severity of oropharyngeal dysphagia. The authors made the items as clear and accurate as possible by consulting with these experts and patients and rewording items as necessary.

All of this work was performed following the Delphi method with respect to such crucial aspects as anonymity, iteration (i.e., executing the procedure in a series of rounds with the exception of the HRQOL questionnaire, which was subjected to just one round), controlled feedback by the facilitators, and statistical group response. Before starting the survey in accordance with the Delphi method, the researchers also specified a cutoff for agreement of at least 70%, which is a valid cutoff for this purpose (any cutoff for agreement greater than 50% is accepted as valid.)

I have only a few suggestions:

Page 7, line 228 to page 8, line 255:

1.      The authors apparently used three levels of agreement: high agreement (agreement≥75%), intermediate agreement (agreement>70% and <75%), and low or lack of agreement (agreement≤70%). If this is correct, I suggest adding a point about this in the subsection of Analysis on Methods. I also suggest that when adding this point, the authors check the consistency in Methods and Results (including Table 3) of the agreement cutoff points and the definition of low or lack of agreement (consensus of at least 70% should be expressed as ≥ 70% and the corresponding expression for low or lack of agreement as agreement < 70%).

2.      The authors should more clearly explain in lines 228 to 255 what was done regarding the items reaching consensus and not reaching consensus. The decisions made regarding these items are somewhat mixed up in the text; particularly, what was done with ten of the original 43 FHS items that did not reach consensus. It seems that eight of these ten were deleted, one because of overlap and the other seven because of agreement <70%. According to Table 3, the other two had an agreement between 70% and 75% and were not deleted but only reworded. These decisions are not clear in the text of lines 228 to 255.

3.      In Table 3, the separations in the Agreement and Decision columns for Rounds I and II are not clear, especially in the row regarding “70<agreement<75%.” The authors should consider placing this cutoff point in one row; e.g., by using an abbreviation for “agreement” and explaining it in the foot-table, increasing the column width, and/or decreasing the font size.

Results:

4.      Please include the items subjected to Rounds I and II and how they were reworded in a supplementary file, stating that these questionnaires are not the final version and should not be used in clinical practice or research until they have been completely developed and validated.

Discussion:

5.      Please add a statement about the limitations of the study with respect to the Delphi method for developing the health-related quality of life questionnaire. This questionnaire was subjected to only one round (Round II), which is one less than the minimum of two recommended by the method.

References:

6.      References 26 and 27 seem incomplete.
